# Detailed Microparticle Analyses Providing Process Relevant Chemical and Microtextural Insights into the Black Mass

**Mickaël Dadé \*, Thomas Wallmach and Odile Laugier**

Eramet Ideas, 78910 Trappes, France; thomas.wallmach@eramet.com (T.W.); odile.laugier@eramet.com (O.L.)
\* Correspondence: mickael.dade@eramet.com

**Abstract:** Eramet uses a combination of physical and hydrometallurgical treatment to recycle lithium-ion batteries. Before hydrometallurgical processing, mechanical treatment is applied to recover the Black Mass which contains nickel, cobalt, manganese and lithium as valuable elements as well as graphite, solvent, plastics, aluminium and copper. To evaluate the suitability for hydrometallurgical recycling, it is essential to analyse the Black Mass chemically but also with respect to size, shape and composition of particles in the Black Mass. The Black Mass of various battery recyclers was investigated by using a combination of SEM/QEMSCAN® analyses. This specific QEMSCAN® database contains 260 subgroups, which comprise major and minor chemical variations of phases. The database was created using millions of point analyses. Major observations are: (1) particles can be micro-texturally characterised and classified with respect to chemical element contents; (2) important textural and chemical particle variations exist in the Black Mass from several origins leading to different levels of quality; (3) elements deleterious to hydrometallurgical processing (i.g. Si, Ca, Ti, Al, Cu and others) are present in well liberated particles; (4) components can be quantified and cathodes active material compositions (LCO, different NMC, NCA, LFP, etc.) that are specific for each battery type can be identified; (5) simulation of further physical mineral processing can optimise Black Mass purity in valuable elements.

**Keywords:** Li-ion battery; black mass; geometallurgy; automated mineralogy; purification; digital simulations; physical separation; liberation; recycling

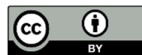

## 1. Introduction

For more than 30 years, many studies have been carried out about the Lithium-ion batteries (LiBs) and their applications [1–3]. In view of the increasing demands of raw materials for LiBs manufacturing, natural resources, such as cobalt (Co) and lithium (Li), will reach a critical level to supply sufficient quantities in the near future [4,5]. The recycling of LiBs is an important part of the larger circular economy concept [6,7]. Existing recycling processes most of LiBs elements to be recovered [7,8].

A cell of LiB is formed of a transition metal compound (active material) as the cathode layer, graphite as the anode layer, aluminium (Al) and copper (Cu) as current collectors, Li salt as the electrolyte, a polymeric separator, organic solvent and a metallic casing [2,3,9,10]. Without changes in the anode foil, the cathode is used to define the obtainable voltage of the LiBs [8]. On the contrary, the anode material, most commonly graphite, should be able to conduct electrons without structural and volume changes [1]. The polymeric separator is placed between electrodes to prevent direct contact and short circuits. This polymeric separator should also be porous enough to allow Li-ion migration [3,10]. The electrolyte consists of a Li salt dissolved in an organic solvent, its main purpose being to efficiently conduct the Li ions between electrodes. PolyVinyliDene Fluoride (PVDF) is used as binder to assemble active material and graphite to the current collector. PVDF should present good mechanical properties as well as good chemical and environmental

endurance [11]. The electric current is carried through Al and Cu foils connected to the cathode and anode, respectively. Finally, the cell components are enclosed in an aluminium (Al), iron (Fe)** or plastic casing [2,3]. At the end of the recycling processes, the composed active material of valuable elements, such as nickel (Ni), manganese (Mn), cobalt (Co) and lithium (Li) can be used to synthesize new precursors for cathodes of future LiBs.

After the pack cells have undergone physical treatments (shredding, grinding, heat treatment, density separation, …), a fine powder, generally called Black Mass (BM), is obtained [9,12]. These BMs consist of fine particles (<500 μm) that are rich in valuable elements such as Ni, Co, Mn and Li [13,14] but also in "impurities" which are deleterious for downstream recycling processes [8,15].

Upstream (physical processing) and downstream (pyrometallurgical, hydrometallurgical processing) processes are generally used to recover valuable elements. At the end of the upstream processes, the composition of BMs has an impact on the performance of recycling downstream processes. To this point, the chemical composition and the presence of major phases must be quantified using several characterisation technics such as SEM [16,17], ICP-EOS [18] or XRD [19,20]. However, these data do not provide enough statistical information on physical properties like particles size, microtextural relationships, as well as liberation and micro-segregation of phases.

In a recent study and for the first time [9], Vanderbruggen et al. have introduced a new analytical methodology to characterize microparticles present in BMs. For this approach, a TESCAN automated mineralogical analyser (TIMA-X), similar to the MLA QEMSCAN® technology [21], was applied. This publication [9] also addressed the importance of BM characterisation during the recycling processes. Three major phases, which are Cu, Al and LiNixMnyCozO (NMC), were identified. This novel approach shows that automated mineralogy analyses is a good method for identifying particles in the BMs. Moreover, it allows us to determine the liberation issue for LiBs components.

In this paper, a refinement of fine (< 500 μm) BM particle chemistries and textures was carried out using an automated mineralogy approach with a high resolution of microanalyses (QEMSCAN® system). In order to develop and calibrate the very detailed database introduced in this paper, three BM samples were characterised to identify and quantify phases and their chemical variations. The second objective of the present paper was to identify a number of additional phases and to quantify those with respect to mass percentages. Moreover, microtextural information, such as the degree of liberation, relative densities and shapes, was quantified for each component. In order to develop and calibrate the very detailed database introduced in this paper, three BM samples were characterised in order to identify and quantify phases and their chemical variations. The second objective of the present paper was to identify a number of additional phases and to quantify those with respect to mass percentages. Moreover, microtextural information such as the degree of liberation and relative densities of phases and particles needed to be quantified. These data are crucial to understanding and monitoring the recovering process. Finally, this paper investigates the potential of automated mineralogical characterisation to purify BMs and attain a lower content of deleterious elements.

## 2. Materials and Analytical Methodology

For this study, three different BMs from different suppliers and recyclers were used. The studied BMs were produced by treatments such as pyrolysis, dry shredding and sieving. In this study, each BM was analysed by means of Inductively Coupled Plasma—Optical Emission Spectrometer (ICP-OES), carbon/sulphur (C/S) analyser and Scanning Electron Microscopy (SEM) coupled with a Quantitative Evaluation of Minerals by Scanning electron microscopy system (QEMSCAN®), which is based at Eramet Ideas (Trappes, France). This system was purchased from ThermoFisher, Eindhoven (Netherland).

## 2.1. Chemical Analyses

The major elements generally found in LiBs were analysed by ICP-OES for each BM. As illustrated in Table 1, the content of Co is higher in the BM 1 compared to the two other BMs. Furthermore, the mass contents of Ni and Mn are of the same order of magnitude (4.70 wt.% and 5.50 wt.%, respectively) but lower than Co (18.00 wt.%). The ratio between Ni, Co and Mn does not indicate the presence of NMC oxide with specific stoichiometry described in the literature (Appendix A). With regard to BM2, the content of Ni, Al and Li are higher than in the other BMs. Moreover, the Ni concentration is approximately three times higher (21.40 wt.%) than the Co and Mn contents (respectively 7.27 wt.% and 7.50 wt.%). For the BM2, the stoichiometry of NMC oxides could be inferred to be 622. Finally, the third BM contains a high concentration of valuable elements, especially Mn (9,32 wt.%). The ratio of the three valuable elements Ni, Co, Mn is close to 2:1:1. Nevertheless, this configuration does not suggest information on the NMC oxide stoichiometry. One essential part of this paper was to identify specific NMC oxide compositions using the QEMSCAN® system.

**Table 1.** Chemical compositions (wt.%) of the three BMs quantified by ICP-OES and C/S analyser.

|        | Ni    | Co    | Mn   | Li   | Al   | Cu   | C     |
| ------ | ----- | ----- | ---- | ---- | ---- | ---- | ----- |
| BM 1   | 4.70  | 18.00 | 5.50 | 3.40 | 5.40 | 4.00 | 54.30 |
| BM 2   | 21.40 | 7.27  | 7.50 | 9.78 | 7.98 | 1.98 | 40.80 |
| BM 3   | 16.40 | 8.33  | 9.32 | 4.25 | 1.11 | 2.15 | 35.38 |

## 2.2. Sample Preparation

The sampling and the representativity of samples are important for automated mineralogical analyses [9]. As described by Vanderbruggen et al. [9], the sampling and the representativity of samples are important for automated mineralogical analyses. In this study, a further aspect of sample preparation has been addressed. The size, mass and density of particles drastically change depending on the phase's nature and microtextural associations. To mitigate the problem of density segregation and get statistically representative data in a two-dimensional mapping, a specific sample preparation protocol was used (Figure 1).

As shown in Figure 1, each BM is embedded in an epoxy resin (a). When the resin has set, the polished section is cut vertically in half (b and c) and embedded one more time into epoxy resin (d and e). This way, the granulometric distribution of BM particles is covered by the polished section. Several polishing steps were performed by means of a Struers RotoForce-3 / RotoPol-31 device using silicon carbide paper < 5µm, then with a diamond polishing suspension on a cloth <1µm size.

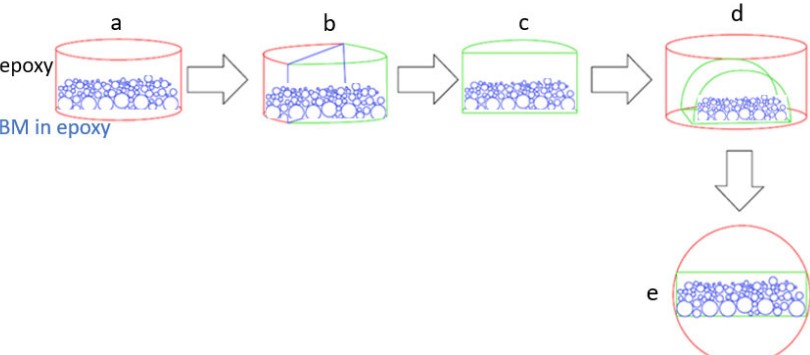

**Figure 1.** Illustration of sample preparation protocol for polished sections. (**a**) Denser particles (large spheres) are more embedded on the bottom of the section then less dense particles (small spheres) ; (**b**) the polished section is cut perpendicular to the flat disc surface; (**c,d**) one half of the

cut disk is again placed into resin producing a second section; (**e**) the polished surface includes all particles without density preferences.

## 2.3. QEMSCAN® Acquisition Parameters

The QEMSCAN® system allows the automation of chemical point analyses on a sample by controlling the SEM stage and the different detectors (BSE and EDS). These measurements must be carried out on polished samples to allow a good quantification.

The electron microscope associated with the QEMSCAN® system (FEI Quanta 650F, ThermoFisher, Waltham, MA, USA) is equipped with a focused electron gun and two EDS and BSE detectors (Bruker, Synergie 4, Lisses, France) of 30 mm². The BSE and EDS detectors are used, respectively, to obtain on each analytical spot a grey level (calibration on standards: quartz = 42, copper = 130, and gold = 232) and a qualitative chemical analysis (calibration on the X-ray spectrum of copper). The aim of the SEM/QEMSCAN® analyses was to investigate, in detail, particles and phase relationships within individual particles that rarely exceed sizes of 100 μm. The QEMSCAN® data were acquired using a FEI Quanta 650 FEG SEM. The system operates with a high voltage of 25 kV, a current of 10 nA and a distance between analytical points between 2.5 and 5 μm.

## 2.4. Brief Introduction to the QEMSCAN® System

The QEMSCAN® associated software includes iMeasure v. 5.4 for data acquisition and iDiscover v. 5.4 for spectral interpretation and data processing.

Using the least squares method, the software associates a simulated spectrum to each experimental EDS spectrum. From this simulated spectrum, the software determines the relative mass concentrations of the elements measured during a spot analysis.

The values obtained for each analysis point (EDS and relative elemental mass concentrations) are compared to a developed database. The latter is made up of different mineral phases. Each of them is defined by a certain number of criteria (elemental concentration and BSE value). Thanks to these successive comparisons, each measurement point is attributed to a mineral phase if it meets the criteria defined for it.

For each phase present in a database, it is possible to enter information, such as the density or the chemical composition of a specific mineral. Given the speed of the EDS measurements of the QEMSCAN® system, data are only qualitative and used to define a mineral, not to measure the exact composition.

The SIP, "Species Identification Profile", is a database from which the information recorded by the SEM is organised and classified according to their chemical composition.

To make the best use of results, the user can choose a pre-existing database or create one. He integrates in the SIP the definitions of the minerals potentially present in the sample. These definitions can be based on an empirical chemical formula called ideal or correspond to chemical analyses from other analytical methods (data from electron microprobe analyser).

Once each measurement point is associated to a mineral phase, it is possible to make many statistical treatments with the QEMSCAN® iDiscover software (Development version 5.4.0.928), for example, particle size distribution curves according to the minerals of interest, calculations of theoretical production yields according to a deposit or an exploitation process. The QEMSCAN® software also allows us to statistically quantify textural data. These data can be used for the evaluation of physical beneficiation possibilities of BMs. To purify BM, it is essential that phases containing valuable elements are not texturally associated with other phases. The liberation concept with respect to free surface of Al foil was well illustrated by Vander-bruggen et al. [9]. In our approach, we use the definition of liberation based on texturally associated phases which will be discussed in greater detail in the following part of this paper. Figure 2 shows examples of both liberation definitions (free surface of a specific phase and the amount of a specific in a particle).

The liberation quantifications presented in the caption of Figure 2 are based on the 2-D sections of the shown particles, and thus, neglect the real 3-D shape of the particle. Only a large number of particles and grains allow an acceptable statistical precision that is

enough to quantify mineral textures. This transformation of 2-D into 3-D information is performed by a stereological algorithm that is included in the iDiscover software.

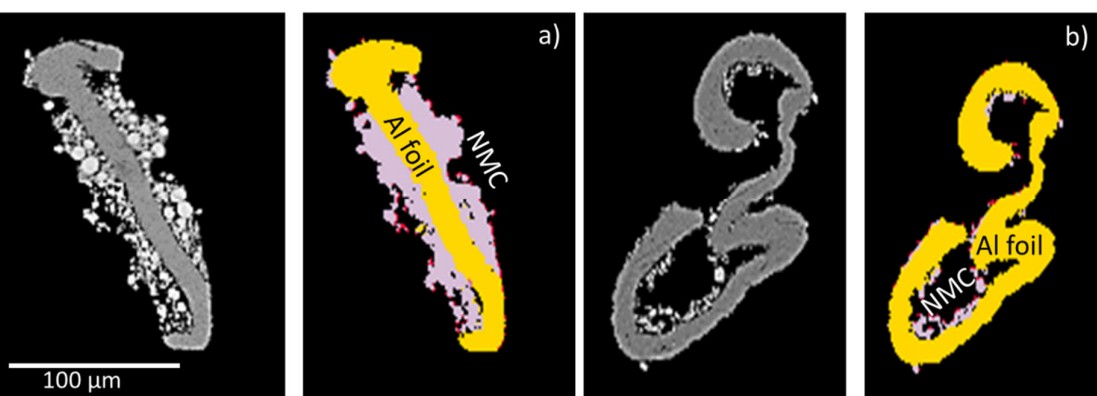

**Figure 2.** Schematic explanation of liberation of Al foil with the BSE images on the left and the QEMSCAN® images on the right. (**a**) Free surface (area percent) of Al foil = 21% and 50 area% (40 mass%) of particle. (**b**) Free surface (area Percent) of Al foil = 67% and 90 area% (82 mass%) of particle. For both images, the red points represent graphite.

### 2.5. Database Development

#### 2.5.1. Statistical Data

To attain a high statistical representativity QEMSCAN® analyses were performed on a total of 3 million particles on which 15.3 million spot analyses were performed. In order to limit the analytical time to less than 3 days per sample, X-ray count rates of 1000 with an average spot of 4 milliseconds/analyse was applied. This yields a high statistical analytical precision which allows us to well define physical and microtextural parameters of particles in samples.

A total of 260 groups with specific chemical variations, defining potentially present phases, are included in the SIP (species identification protocol) file. Each entry of this database is associated with compositional (elemental wt.%) and density (g/cm$^3$) data, which are stored in the so-called primary file.

#### 2.5.2. Methodology

The identification of phases that are present in the investigated BM samples is an essential part for the creation of a comprehensive QEMSCAN® database. This was done by identifying phases based on EDX spectra, which are specific for each phase. Some examples are presented to demonstrate the procedure. These examples include NMC532 ($LiNi_{0.5}Mn_{0.3}Co_{0.2}O_2$), NMC622 ($LiNi_{0.6}Mn_{0.2}Co_{0.2}O_2$), LCO ($LiCoO_2$) and NCA ($LiNi_{0.80}Co_{0.15}Al_{0.05}O_2$).

Each micro analysis, of which the EDX spectrum is stored and available for future verification, was grouped into categories of specific chemical compositions representing one distinct and largely homogeneous phase.

The initial database development started with the identification of groups of distinct phases that contained Co, Ni and Mn. Other elements that are present in LiBs also need to be considered. These elements correspond to Cu and Al foils and other active matters, e.g., LCO and NCA (Appendix A).

These groups representing the major phases of current industrial LiBs were refined into subgroups. For example, the various proportions of Ni, Mn and Co allow us to classify them into several groups of NMCs depending on the stoichiometry. Figure 3 present EDX spectra of NMC532 and NMC622 acquired during QEMSCAN® analyses. In this example, the various peak intensity and the concentration ratio of Mn and Ni allow us to

distinguish these two types of NMCs. Furthermore, the NMC532 and NMC622 analyses compare very well to ideal spectra. Moreover, the absence of oxygen in some EDX analyses of NMCs were used to identify metal subgroups.

Apart from NMC532 and NMC622, other EDX spectra of the most industrially used active matters such as LCO, NCA and LFP (LiFePO4) are presented in Figure 3c to Figure 3e. Once again, a good agreement between ideal spectra and QEMSCAN® generated spectra were obtained.

With the same approach, the Al group was separated into metal and oxide subgroups. Silicates including Si, K, Al, Fe and Mg were also identified and need to be represented and grouped into the QEMSCAN® database.

It is evident that a short analytical acquisition time, which is a prerequisite for the acquisition of high quantities of analyses (several million of analytical spots per sample), provides sufficient information to distinguish all phases in each BM studied. Photon counts and their relationships are used as entries for the phase identification database.

All the entries can then be grouped in larger compositional groups allowing for a visualisation and quantification of phases, phase compositions and textural associations.

It needs to be noted that the Li, due to its low atomic number, is not measured by SEM analyses and the resulting EDX spectra. In the present study, this element can still be quantified by inference. This will be addressed in the detailed description of the characterized BMs.

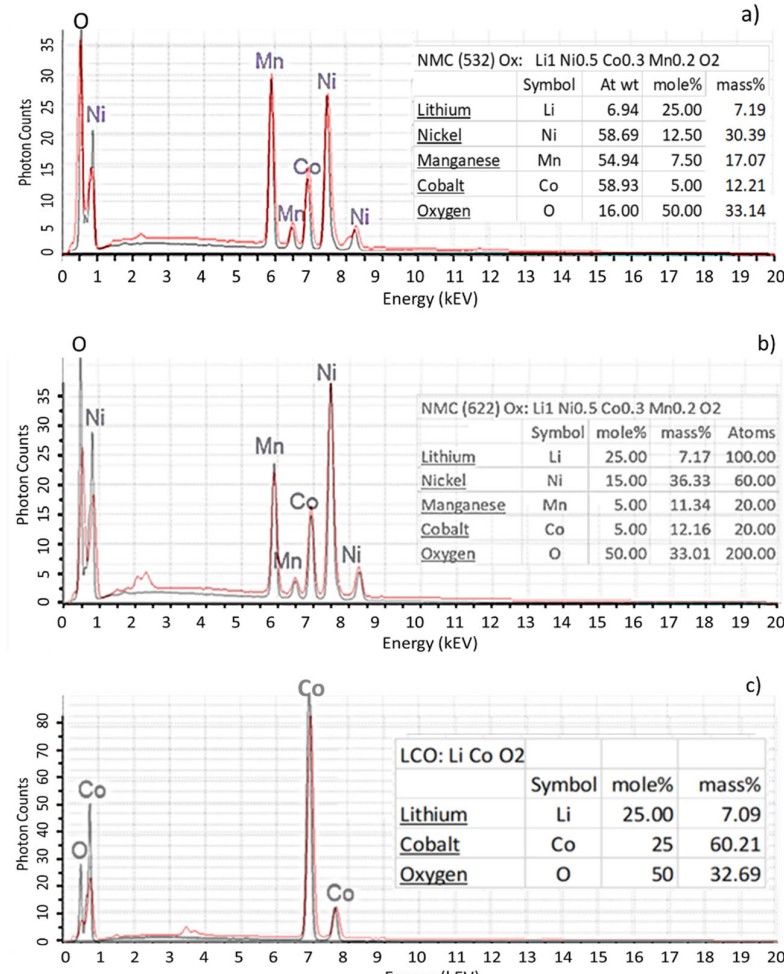

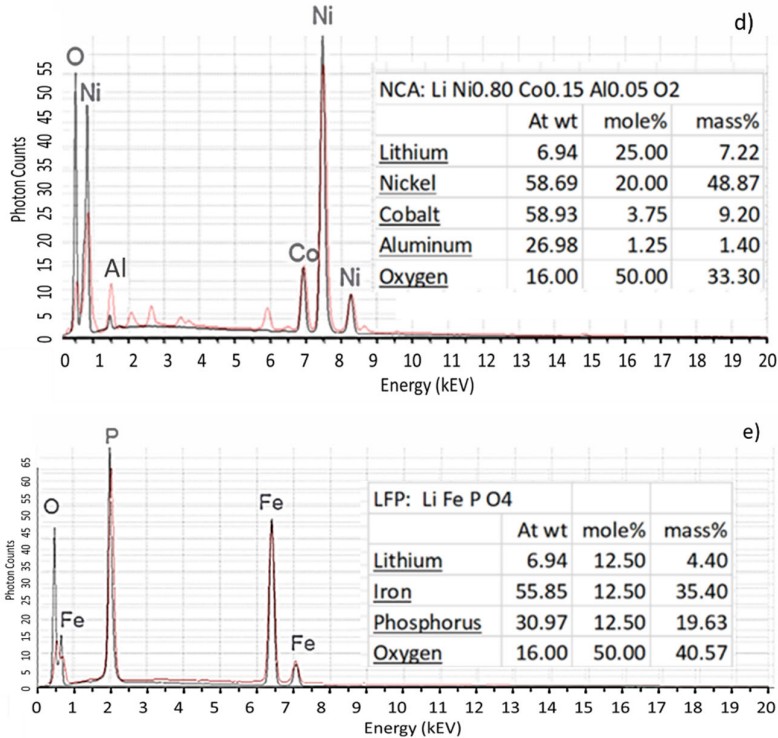

**Figure 3.** Spectra of NMC 532 (**a**), NMC 622 (**b**), LCO (**c**), NCA (**d**) and LFP (**e**). Red lines represent QEMSCAN® EDX spectra; black lines correspond to modelled EDX spectra using the QEM-SCAN® iDiscover software; At wt: atomic weight of element; mole%: composition of phase ex-pressed in moles; mass%: composition of phase expressed in elemental weight present.

### 2.5.3. Database Verification

Once a database is generated, based on the identification of phases with various major and minor compositional differences, the validity must be tested.

One method of testing the quality of our QEMSCAN® database is performed by comparing the recalculated sample compositions with the results from chemical analyses such as ICP-OES analyses. This comparison can only be performed if the density of each phase is defined in the QEMSCAN® database. Element concentrations in dense phases result in higher element concentrations than in less dense phases.

No comprehensive dataset on densities of the identified phases is described in the literature. It was necessary to identify relative densities of all identified phases to recalculate element concentrations in the studied BMs. By iteratively adjusting phase densities for all phases included in the QEMSCAN® database using compositionally variable BMs, it was possible to obtain recalculated chemical sample compositions and compared to ICP-OES analyses (Table 2). In the literature [9], a first approach was carried out with a logarithmic correlation. In this present study, a direct non-logarithmic element comparison between ICP-OES and QEMSCAN® analyses was preferred to show a higher precision in the correlation. As illustrated in Figure 4a, QEMSCAN® analyses match very well with ICP-OES generated analyses ($r^2 > 0.99$).

The good correlation of QEMSCAN® chemical assays with ICP-OES generated analyses (Table 1) needs to be further verified by improving and testing the developed database on many more BMs with variable chemical and phase compositions. In this case, Figure 4b,c illustrate the validation of QEMSCAN® database by a good agreement between QEMSCAN® and ICP-OES results. Three other BMs, which are not presented in this study, were not used to define the densities but only to further validate the QEMSCAN® database.

As already mentioned above, Li and C cannot be measured during SEM/ QEM-SCAN® analyses and need to be analysed externally by using ICP-OES and a C/S analyser. The C content in the sample is presented in Table 1. The corrected weight percent of phases after adding the C concentration will ideally result in a good assay reconciliation. Using the stoichiometric compositions for activated phases, including their Li content, yields a good correlation; it also yields a good correlation with Li.

**Table 2.** Comparison between ICP-OES chemical compositions (wt.%) of the three BMs and recalculated values using QEMSCAN® database.

| | Assay | Ni | Co | Mn | Li | Al | Cu | Fe | Ti | Si | Ca | Mg |
|---|---|---|---|---|---|---|---|---|---|---|---|---|
| BM 1 | Chem | 4.70 | 18.00 | 5.50 | 3.40 | 5.40 | 4.00 | 1.80 | - | 2.20 | 0.15 | 0.04 |
| (a) | QEM | 4.38 | 17.80 | 5.66 | 3.40 | 5.41 | 3.90 | 1.43 | - | 2.03 | 0.06 | 0.12 |
| BM 2 | Chem | 21.40 | 7.27 | 7.50 | 9.78 | 7.98 | 1.98 | 0.09 | 4.00 | - | - | - |
| (b) | QEM | 21.39 | 7.31 | 7.24 | 9.67 | 7.76 | 2.29 | 0.38 | 3.82 | - | - | - |
| BM 3 | Chem | 16.40 | 8.33 | 9.32 | 4.25 | 1.11 | 2.15 | - | - | - | - | - |
| (c) | QEM | 16.57 | 8.35 | 9.46 | 4.53 | 1.26 | 2.07 | - | - | - | - | - |

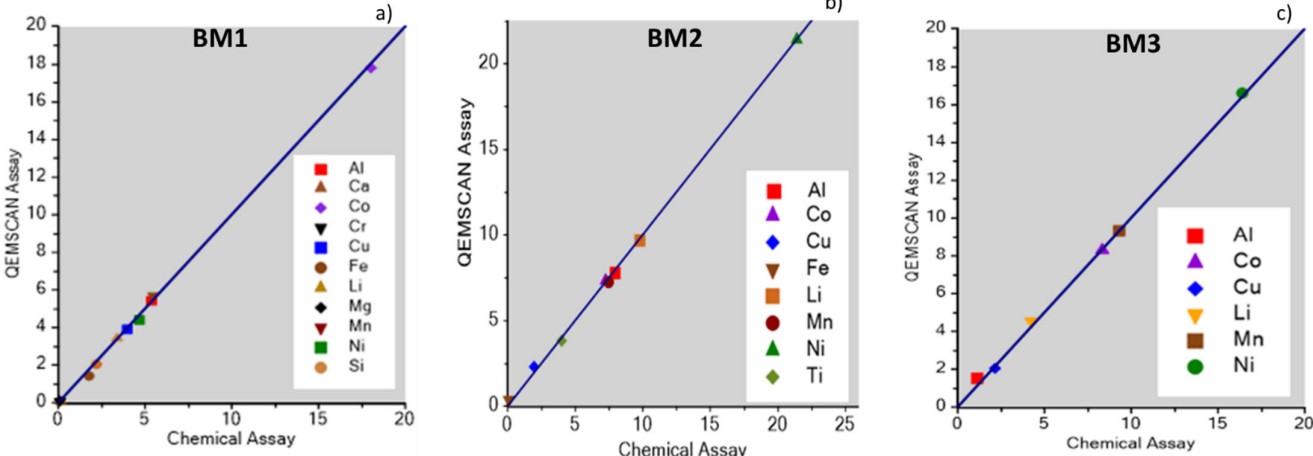

**Figure 4.** Graphical comparison of ICP-OES analyses with QEMSCAN® derived element concentrations for BM 1 (**a**), BM 2 (**b**) and BM 3 (**c**). The blue lines correspond to a correlation coefficient r2 equivalent to 1.

## 3. Micro-Texture and Chemistry of Particles

### 3.1. Particle Size Distributions

One main difference between the three analysed BMs is demonstrated in the particle size distributions. The QEMSCAN® software provides a good comparison between the three samples. An advantage of the QEMSCAN® software is the application of a stereological algorithm that converts 2-D images into 3-D information. This was tested and shown to provide reliable results, provided that a large quantity of particles is analysed [22,23].

Particle size distributions of the three analysed BMs are presented in Figure 5. They are distinctly different with BM 1 consisting of overall coarser particles than BM2. BM3 contains particles that are much smaller than those of BM1 and BM2.

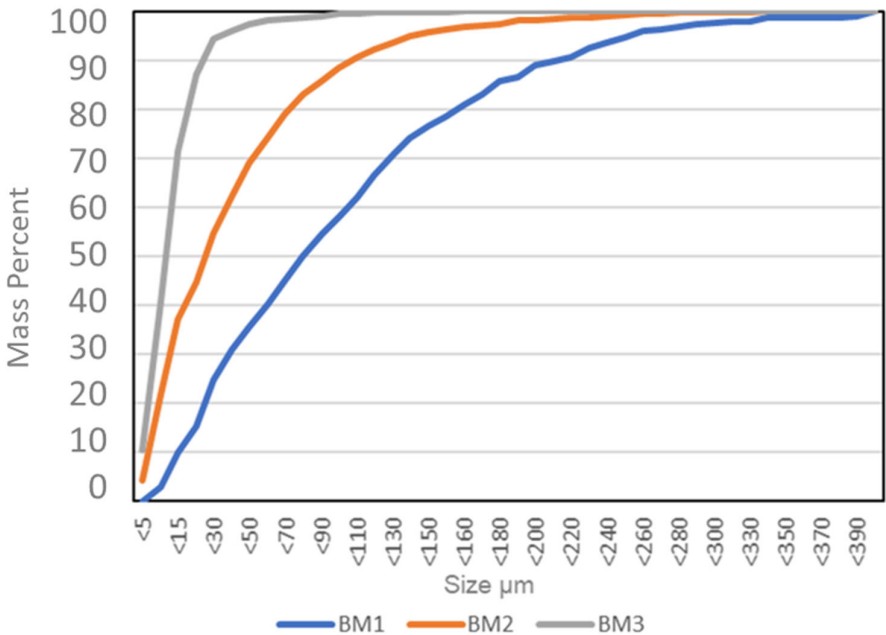

**Figure 5.** Particle size distribution of the three analyses of BMs. The particle size distributions are clearly different.

### 3.2. *Phase Compositions and Textural Descriptions of BMs*

### 3.2.1. BM1: Composition and Textural Information

The most variable of the characterised BMs is BM1 illustrated in Figure 6. The characterisation was performed on 10.5 million X-ray analyses using 5 μm steps on a total of 753,000 particles. In the right part of Figure 6, the chemical and phase composition of BM1 is presented. The particularity of this BM is that NMC phases are not abundant. LCO was the most prominent of active matter still associated principally with Al oxide. This association, corresponding to the initial cathodes of cell phones, shows the weak liberation for the Co.

Other active matter as LNO (LiNiO) and LMO (LiMnO) were indexed and visualized in Figure 6.

Cu foils as well as rounded particles corresponding to chlorides of Ni and Co were also identified.

This BM could generate problems during hydrometallurgical processing. One reason for the hydrometallurgical complications that could be encountered is the presence of deleterious impurities such as silicates and Fe oxides. As illustrated in Figure 6, the silicates were identified and belong to commonly occurring rock forming minerals, including quartz, K-feldspar, pyrophyllite and chlorite. Compared to Fe oxides impurities, silicates come from more external sources than LiBs.

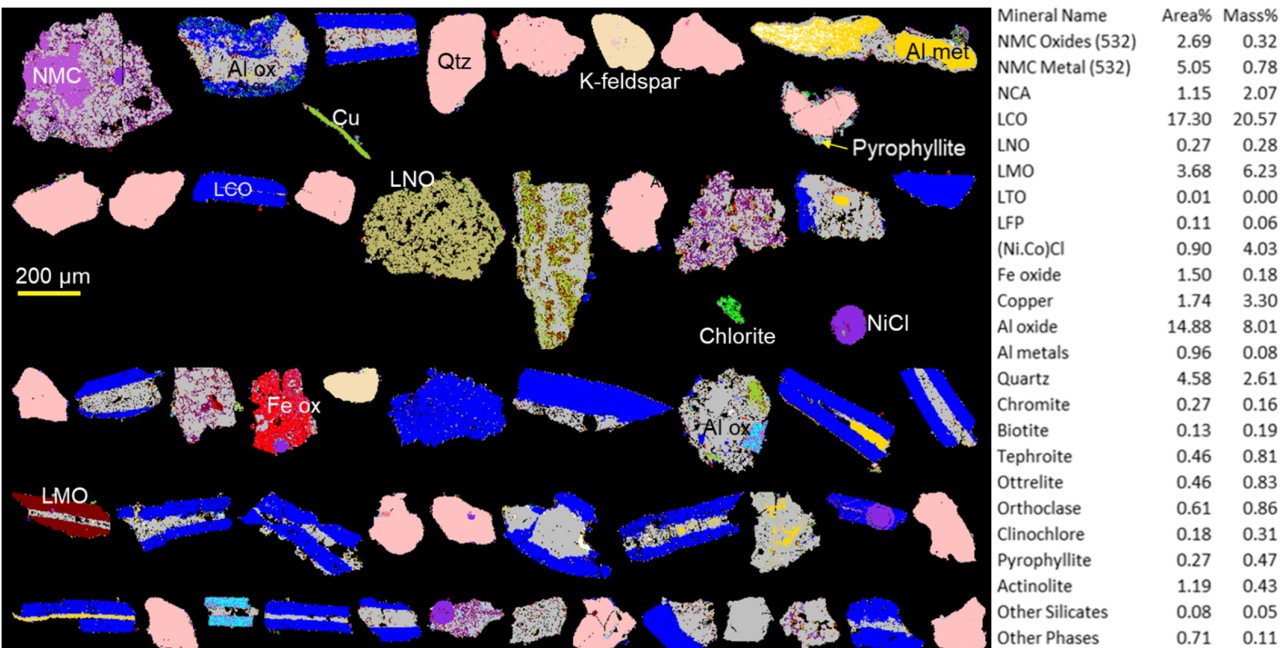

**Figure 6.** Selected QEMSCAN® image showing activated phases (NMC, LMO) and silicates such as quartz (Qtz), pyrophyllite and chlorite (**left**). Also shown are Fe oxide, Al oxide, Al metal, NiCl and Cu. The QEMSCAN® element compositions are recalculated in mass percent, which differ from the area percent in the sample (**right**).

### 3.2.2. BM2: Composition and Textural Information

Figure 7 illustrates some textural relationships between the phases present in the sample BM2.

The characterisation was performed on 74.2 million X-ray analyses using 5 μm steps on a total of 1.6 million particles. The composition of BM2 differs from BM1. Most of this sample consists of NMC phases and Al oxides. Several NMC compositions (NMC622, NMC532 and NMC811) were identified in this BM. These NMC phases have large variations in O and were classified in NMC oxides and NMC metals. Alongside NMCs, a variety of other activated phases were identified. Theses phases are NCA, LCO, LNO, LMO, LTO and LFP. QEMSCAN® measurements show that some Al oxides contain Ti, which warrant the inclusion of a specific database entry (Al doped LTO). The concentrations of silicates are low.

Spherical, Cu and Al metal particles can be noted in Figure 7. This type of particles suggests a pyrolysis process used to recycle the initial batteries.

Some analyses show only F or P as elements. As F is not stable as a specific solid phase it was assumed that Li, that cannot be measured by means of SEM, is associated with F representing the LiF obtained after LiPF$_6$ decomposition. Further investigations are required to prove this hypothesis.

Chemical and phase compositions of sample BM2 are shown in the right part of Figure 7. A selected QEMSCAN® image of phases larger than 100 μm is presented in Figure 7.

Figure 8 shows a detailed QEMSCAN® image of a particle that is variable in composition. Most of this particle represents Al metal but areas rich in NMC metal, AlMn oxide, CoNi metal, Mn and Ti can be observed. The element Ti can occur as Ti oxide or as a mixture of Ti oxide and Al oxide. This particle exemplifies well the intricate task of developing a comprehensive database that takes into account the large variability of phases.

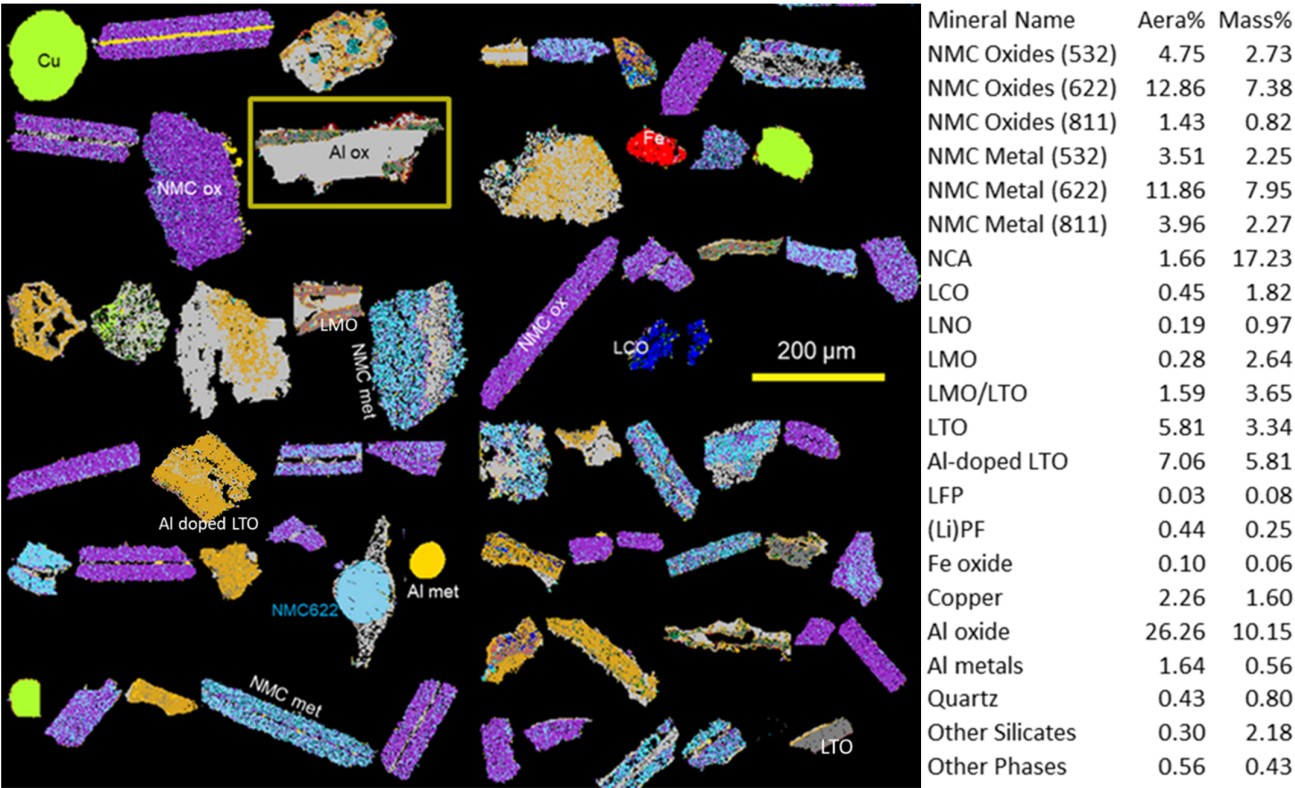

| Mineral Name | Aera% | Mass% |
|---|---|---|
| NMC Oxides (532) | 4.75 | 2.73 |
| NMC Oxides (622) | 12.86 | 7.38 |
| NMC Oxides (811) | 1.43 | 0.82 |
| NMC Metal (532) | 3.51 | 2.25 |
| NMC Metal (622) | 11.86 | 7.95 |
| NMC Metal (811) | 3.96 | 2.27 |
| NCA | 1.66 | 17.23 |
| LCO | 0.45 | 1.82 |
| LNO | 0.19 | 0.97 |
| LMO | 0.28 | 2.64 |
| LMO/LTO | 1.59 | 3.65 |
| LTO | 5.81 | 3.34 |
| Al-doped LTO | 7.06 | 5.81 |
| LFP | 0.03 | 0.08 |
| (Li)PF | 0.44 | 0.25 |
| Fe oxide | 0.10 | 0.06 |
| Copper | 2.26 | 1.60 |
| Al oxide | 26.26 | 10.15 |
| Al metals | 1.64 | 0.56 |
| Quartz | 0.43 | 0.80 |
| Other Silicates | 0.30 | 2.18 |
| Other Phases | 0.56 | 0.43 |

**Figure 7.** Selected QEMSCAN® image showing activated phases (NMC, LCO) Also shown oxide, Al oxide, Al metal and Cu. Cu and NMC are rounded, indicating partial melting of the material during pyrolysis (**left**). Highlighted particle (frame) is shown in greater detail in Figure 8. The QEMSCAN® element compositions are recalculated on the mass percentages of phases, which differ from their area percent distribution in the sample (**right**).

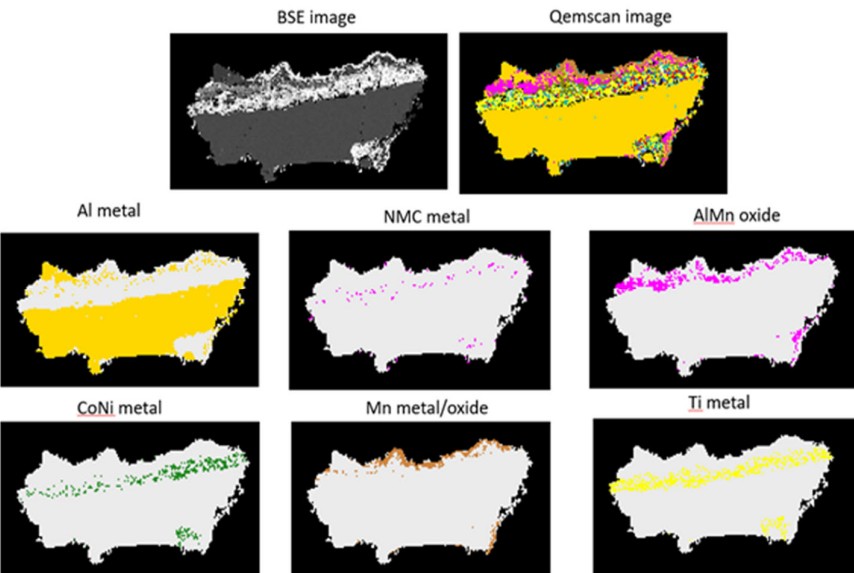

**Figure 8.** QEMSCAN® analyses allow to distinguish between different phases in individual particles. An example is shown in the above figure.

### 3.2.3. BM3: Composition and Textural Information

The characterisation was performed on 11.8 million X-ray analyses using 5 μm steps on a total of 1.4 million particles. The composition of BM3 differs from BM1 and BM2 in that this sample is very rich in Ni. The right part of Figure 9 shows that most of this sample consists of NMC532 oxides with no other NMC compositions. Only minor amounts of other activated phases and silicates are present.

A selected QEMSCAN® image of phases larger than 100 μm showing the textural relationships between Al oxide, Al metal, Cu and NMC532 is presented in Figure 9. A large part of NMC oxides is separated from their Al foils. Moreover, Al foils are rolled up as well as Cu foils. This suggests that a milling was used to separate active matter from Al foils.

To use and quantify these and other textural features, the QEMSCAN® integrated modelling option can be applied.

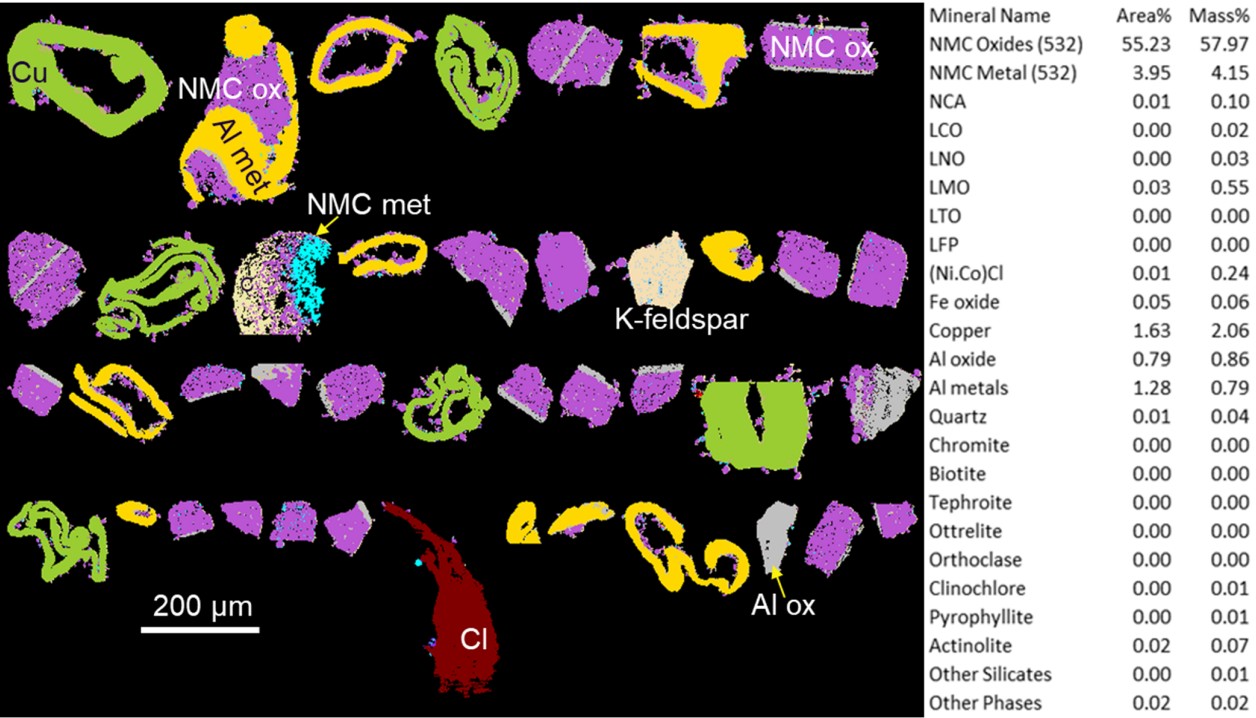

| Mineral Name | Area% | Mass% |
|---|---|---|
| NMC Oxides (532) | 55.23 | 57.97 |
| NMC Metal (532) | 3.95 | 4.15 |
| NCA | 0.01 | 0.10 |
| LCO | 0.00 | 0.02 |
| LNO | 0.00 | 0.03 |
| LMO | 0.03 | 0.55 |
| LTO | 0.00 | 0.00 |
| LFP | 0.00 | 0.00 |
| (Ni.Co)Cl | 0.01 | 0.24 |
| Fe oxide | 0.05 | 0.06 |
| Copper | 1.63 | 2.06 |
| Al oxide | 0.79 | 0.86 |
| Al metals | 1.28 | 0.79 |
| Quartz | 0.01 | 0.04 |
| Chromite | 0.00 | 0.00 |
| Biotite | 0.00 | 0.00 |
| Tephroite | 0.00 | 0.00 |
| Ottrelite | 0.00 | 0.00 |
| Orthoclase | 0.00 | 0.00 |
| Clinochlore | 0.00 | 0.01 |
| Pyrophyllite | 0.00 | 0.01 |
| Actinolite | 0.02 | 0.07 |
| Other Silicates | 0.00 | 0.01 |
| Other Phases | 0.02 | 0.02 |

**Figure 9.** Selected QEMSCAN® image showing the largest particles in the sample including NMC532, Al oxide, Al metal and Cu. Cu and NMC are rounded, indicating partial melting of the material during pyrolysis (**left**). A particle containing only Cl could be LiCl, which needs to be verified. The QEMSCAN® element compositions are recalculated on the mass percentages of phases (**right**).

## 4. Digital Simulations: A Geometallurgic Approach

One of many uses of geometallurgy is the prediction of mineral and particle behaviours during the physical processing of ores. Differences in size, density, magnetic susceptibilities of particles and grains rank high in the list of properties that can be used in digital modelling to identify the most adequate beneficiation method or a combination of methods to enrich valuable elements hosted in ores. The QEMSCAN® software allows geometallurgical modelling by using the EDX acquired data in combination with a database.

As spent LIBs are rich in valuable metals, these materials can be regarded as secondary high-grade ores made by human activities. The resulting BMs are intermediary and enriched materials that need to be further processed, up to hydrometallurgical processes in order to recover battery grade materials.

Some selected examples using particle size and densities of sample BM3 are presented in the following chapters.

Geometallurgical concepts can be applied to enrich BMs before hydrometallurgical processing takes place. This potentially reduces the mass of material to be processed, and thereby the amount of chemicals and energy. This bears a potential advantage to reduce the environmental impact.

In this study, the BM3 was used to apply this geometallurgy approach for purification matters using physical properties and textural information.

### 4.1. Liberation of Phases in the Original BM3

One advantage of QEMSCAN® analyses is that many textural features can be quantified. One textural feature of importance is the liberation. Figure 10 shows the liberation and textural association of the most prominent phases identified in sample BM3. The liberation of Cu with 72 wt.% liberated at more than 80% is good and warrants further considerations to remove this phase using physical processing. The overall liberation of NMC is lower, and the liberation of Al oxides and metals is poor. Aluminium is often attached to NMC particles.

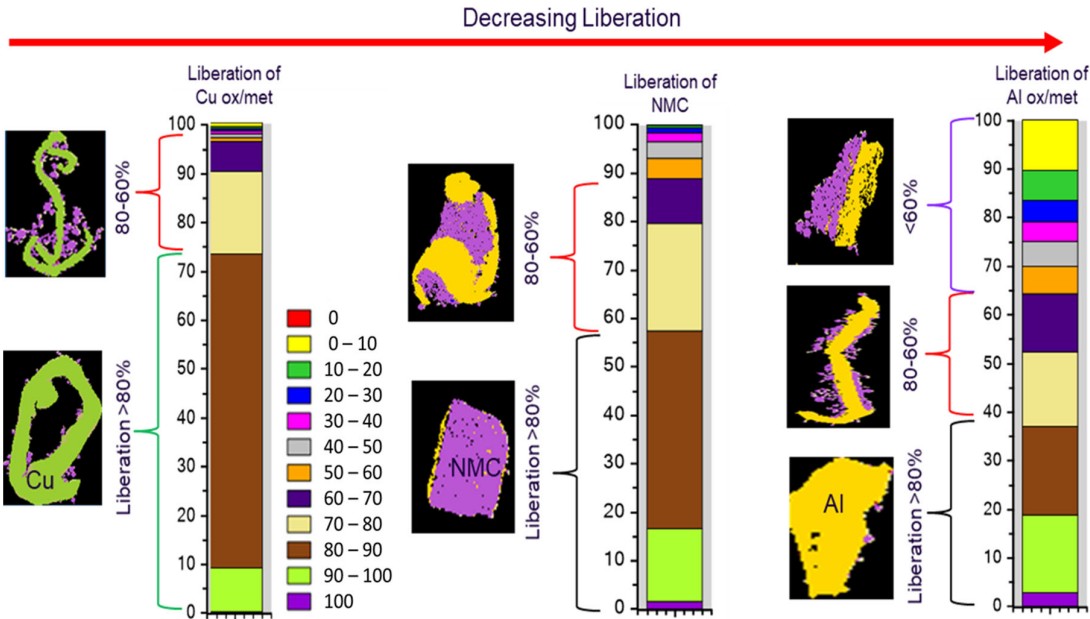

**Figure 10.** QEMSCAN® generated liberation percentages of Cu oxides and metals, NMC and Al oxides and metals in sample BM3. The indicated brackets represent the average liberation of all analysed particles. For each column, the coloration expresses the mass% liberation.

It should be noted that it is of great importance to statistically evaluate the liberation of phases containing elements of value, as this is a prerequisite to consider physical processing.

### 4.2. Density Parameters to Recover Metals from BM through Physical Methods

As the QEMSCAN® database includes density information, a simulated density separation was performed. QEMSCAN®-based digital modelling results are presented Figure 11. Cu, Al and NMC phases are expressed in terms of relative density and mass percent according to QEMSCAN®. The mass percentage is normalized to 100% to compare differences. Copper phases are generally denser than NMC phases. Aluminium phases are less dense than Cu and NMC phases. The results show that NMC phases could be enriched, and Cu phases diminished with a density separation at a density threshold of 5.

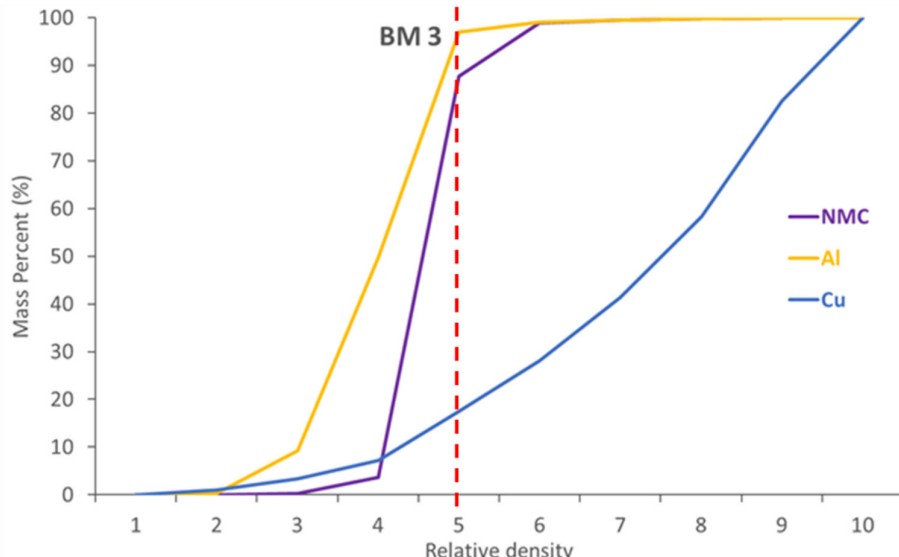

**Figure 11.** QEMSCAN® generated relative density vs mass modelling for Cu, Al and NMC phases. The red dashed line refers to a density cut-off.

*4.3. Particle Size Parameters to Recover Metals from BM through Physical Methods*

Digital simulation can also be performed using granulo-mineralogical or granulo-chemical modelling shown in Figure 12 and Figure 13, respectively. With decreasing particle sizes, the amount of Al and Cu diminishes, whereas Co, Ni and Mn (NMC) contents increase. Figures 12, 13 and 14 show that fine grained particles (<20 μm) contain very small amounts of Cu (≈ 3 mass%) and reduced amounts of Al (≈ 37 mass%). On the contrary, 97 mass% of NMC from the initial BM3 are contained in the fraction <20 μm (Figure 14). Relatively large particles (>20 μm) contain mainly Al and Cu.

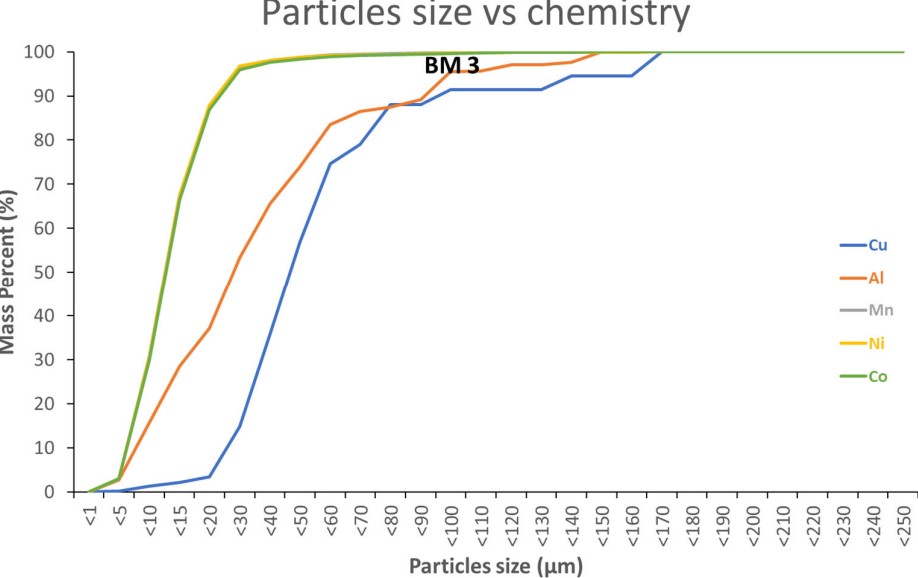

**Figure 12.** QEMSCAN® generated granulo-chemical modelling. In the BM3 sample Cu and Al increase in abundance with increasing particle sizes. This information can be used to increase the Co and Ni concentrations by particle size fractionation. The particle size distribution of BM3 is shown in Figure 5.

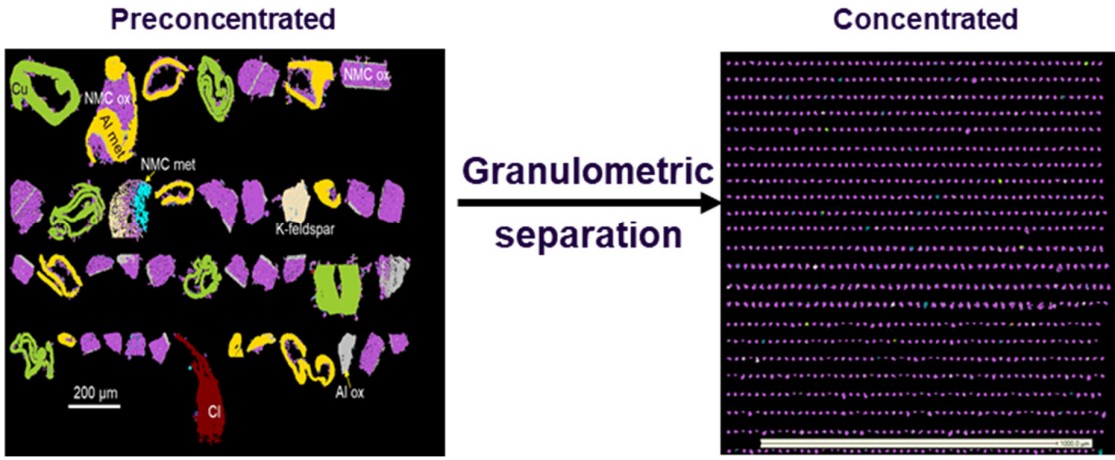

**Figure 13.** QEMSCAN® images showing large particles typically enriched in Cu and Al (>20 μm), and a very small sized fraction (<20 μm) enriched in NMC with few Cu and Al particles.

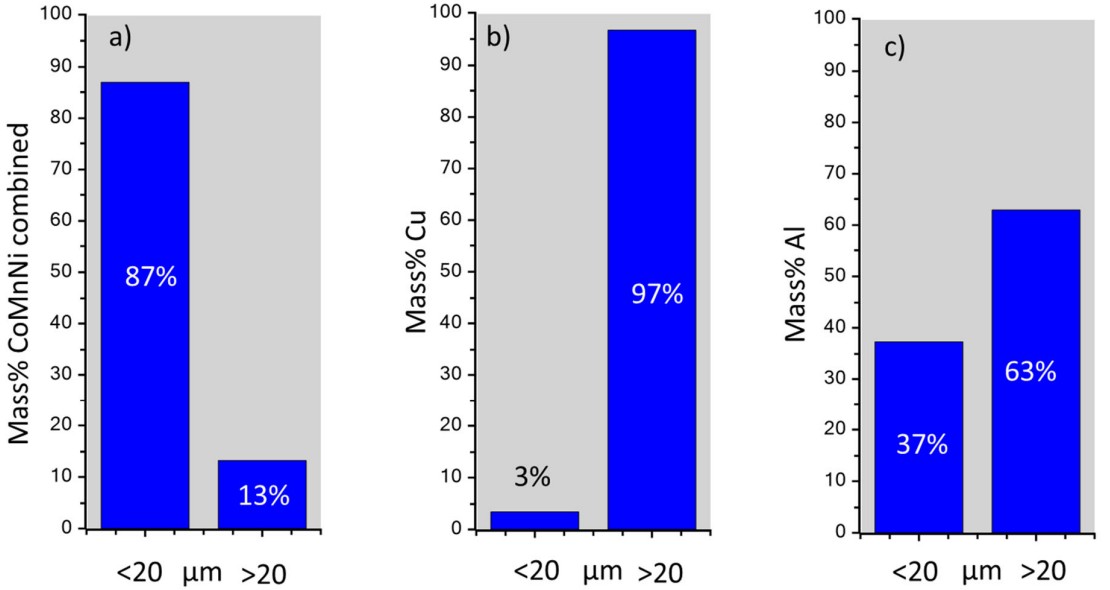

**Figure 14.** Simulated repartition in mass% of NMC (**a**), Cu (**b**) and Al (**c**) using a particle size cut-off of 20 μm.

It needs to be noted that modelling was performed using the developed prototype database, which still has room for improvement. The modelling result can also vary in BMs of different compositions.

These preliminary results show the importance of simulation for the pre-processing of BMs to physically remove some of the deleterious elements, such as Al and Cu, before hydrometallurgical or pyrometallurgical processing.

The current paper is not dedicated to process modelling, which requires more extensive discussions and simulation verifications by means of physical testing. These results will be presented in a future publication.

## 5. Conclusions and Perspectives

This study illustrates the necessity to characterise BMs provided by supplier or recyclers to increase their purity before the hydrometallurgical process. Advanced characterisation techniques, such as an automated mineralogy system coupled with a SEM, provides quantitative microtextural information for each BM particle. This information is crucial to understand and prevent potential difficulties.

For the first time, a prototype of a QEMSCAN® database was implemented to categorise each component of LiB as well as their stochiometric and chemical variations. The different types of NMC were distinguished as well as the oxidation state of aluminium. In addition, it has been observed that BMs contain a mixture of several LiBs active matter such as LCO, NMC, NCA, LFP and so on. All these categorisers demonstrate the robustness of the QEMSCAN® database, which can potentially be used for every type of LiBs.

Digital modelling, using the QEMSCAN® integrated software, was performed. The digital modelling approach is very similar to that commonly used in the geometallurgical modelling of natural ores. By using information on liberation, particle sizes and particle densities of phases hosted by BMs, it is potentially possible to apply physical beneficiation steps before hydrometallurgical processing of BMs in order to pre-enrich BM samples with valuable elements such as Co, Ni and Li.

This should be further considered to render hydrometallurgical processing more economic, in terms of mass to be treated and acid consumption, and therefore more respectful of the environment.

In this paper, graphite and plastics were acquired from external analyses (carbon/sulphur analyser). For the future, it is of interest to quantify these two components to complete the QEMSCAN® database.

Several studies have shown the difficulty to distinguish the graphite from the epoxy resin [9,24,25]. First tests show that this is possible using the QEMSCAN® system (Figure 15). EDX analyses show very specific compositions of graphite and epoxy resin. These variations are used to distinguish carbon compounds. Moreover, plastic cells and membrane separators can also be indexed. The quantification of these components will be crucial in determining the recycling efficiency but also in resolving analytical problems with the differentiation of carbon content on plastics and graphite.

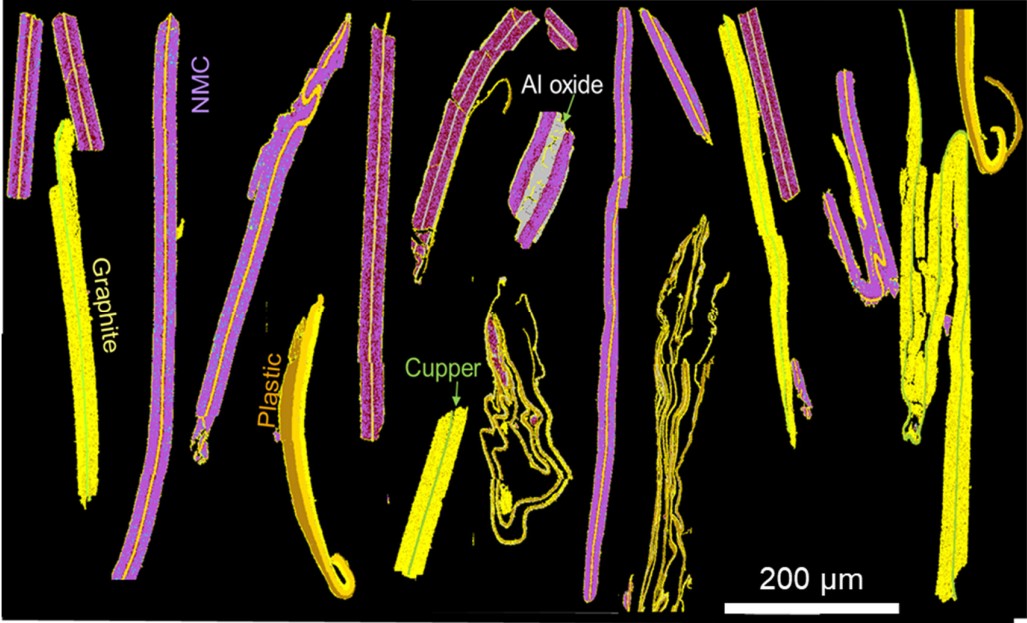

**Figure 15.** Indexation and distinction of graphite (in yellow), plastics (in brown) and the epoxy resin (in black).

**Author Contributions:** Investigation, M.D. and O.L.; Methodology, M.D., T.W. and O.L.; Resources, T.W. and O.L.; Software, M.D. and T.W.; Supervision, M.D.; Validation, M.D. and T.W. All authors have read and agreed to the published version of the manuscript.

**Funding:** This research received no external funding.

**Data Availability Statement:** Not applicable

**Acknowledgments:** Acknowledgments: Authors would like to acknowledge Gabriel CRUMIERE, Arthur PERROTON and Marion PIERRE for the chance to use and study Black Mass, and Laurent DELPEUCH for the ICP-OES analyses. We also thank the anonymous reviewers for their constructive comments.

**Conflicts of Interest:** The authors declare no conflict of interest.

## Appendix A

The theoretical chemical composition according to the stoichiometry of several active matters is given in the following table.

**Table A1**. Summary of theoretical chemical composition (wt%) of active material indexed in this paper.

| | Formula | Theoretical Chemical Composition (wt.%) | | | | | | | | |
|---|---|---|---|---|---|---|---|---|---|---|
| | | Li | Ni | Mn | Co | Al | Ti | Fe | P | O |
| LCO | $LiCoO_2$ | 7.09 | - | - | 60.21 | - | - | - | - | 32.69 |
| NMC111 | $LiNi_{0.33}Mn_{0.33}Co_{0.33}O_2$ | 7.24 | 20.20 | 18.91 | 20.28 | - | - | - | - | 33.37 |
| NMC442 | $LiNi_{0.4}Mn_{0.4}Co_{0.2}O_2$ | 7.22 | 24.41 | 22.85 | 12.25 | - | - | - | - | 33.14 |
| NMC532 | $LiNi_{0.5}Mn_{0.3}Co_{0.2}O_2$ | 7.19 | 30.39 | 17.07 | 12.21 | - | - | - | - | 33.14 |
| NMC622 | $LiNi_{0.6}Mn_{0.2}Co_{0.2}O_2$ | 7.17 | 36.33 | 11.34 | 12.16 | - | - | - | - | 33.01 |
| NMC811 | $LiNi_{0.8}Mn_{0.1}Co_{0.1}O_2$ | 7.13 | 48.27 | 5.65 | 6.06 | - | - | - | - | 32.89 |
| NCA | $LiNi_{0.8}Co_{0.15}Al_{0.05}O_2$ | 7.22 | 48.87 | - | 9.20 | 1.40 | - | - | - | 33.30 |
| LNO | $LiNiO_2$ | 7.11 | 60.12 | - | - | - | - | - | - | 32.77 |
| LMO | $LiMnO_2$ | 7.39 | - | 58.52 | - | - | - | - | - | 34.09 |
| LFP | $LiFePO_4$ | 4.40 | - | - | - | - | - | 35.40 | 19.63 | 40.57 |
| LTO | $LiTiO_2$ | 8.00 | - | - | - | - | 55.14 | - | - | 36.86 |

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
