# Peer review of "Detailed Microparticle Analyses Providing Process Relevant Chemical and Microtextural Insights into the Black Mass"

_minerals, doi:10.3390/min12020119_

Round 1
Reviewer 1 Report
The authors wrote the manuscript on “Detailed microparticle analyses providing process relevant chemical and microtextural insights into the black mass”. The results obtained by the authors are excellent. However, there is lack of explanation with respect to actual analysis in the manuscript.
The authors should clarify before publishing. Some minor observations are:
- The chemical composition (wt %) given in table 1 is not 100%. This means some more constituent is present in the sample apart from Ni, Co, Mn, Li, Al, Cu, and C.
- 1 should be explained as a, b, c, d, e and f with captions.
- In fig. 2, it should be “schematic diagram of”…instead of “example of”…
- Why are the authors got only one grain in fig. 2? Give the mechanism.
- The both pictures given in fig. 3, 4, 5, 6, and 7 are almost same. As the authors were comparing, it should be merged in one graph. The font of X and Y-axis is very small. Why the total At. % is not 100?
- Particle size distribution was shown in fig. 9. However, if it can be compared with the particle size analyzer test, it will be more appropriate.
- The composition and textural information is looks interesting. However, if pure metal, alloy or HEA is forming, can it be analyzed?
- How about the gas detection (for example O2) using this methodology?
Author Response
The authors wrote the manuscript on “Detailed microparticle analyses providing process relevant chemical and microtextural insights into the black mass”. The results obtained by the authors are excellent. However, there is lack of explanation with respect to actual analysis in the manuscript.
The authors should clarify before publishing. Some minor observations are:
- The chemical composition (wt %) given in table 1 is not 100%. This means some more constituent is present in the sample apart from Ni, Co, Mn, Li, Al, Cu, and C.
As said in line 90, table1 just give the major elements of LiB. Other elements as Si, Fe and O are also present in minor amount which explains why the total do not attain 100%.
- 1 should be explained as a, b, c, d, e and f with captions.
Figure 1 was modified following the suggestion of the reviewer
- In fig. 2, it should be “schematic diagram of”…instead of “example of”…
Authors have modified the text with “Schematic explanation of ...”
- Why are the authors got only one grain in fig. 2? Give the mechanism.
Fig.2 allows to introduce the principe of liberation and more precisely the liberation degree.
- The both pictures given in fig. 3, 4, 5, 6, and 7 are almost same. As the authors were comparing, it should be merged in one graph. The font of X and Y-axis is very small. Why the total At. % is not 100?
In fig. 3, 4, 5, 6 and 7, we compare the QEMSCAN spectra with the ideal spectra simulated by QEMSCAN software. We do not search to compare all phases just to compare acquired and simulated spectre. However, it is impossible to superpose both spectra with the QEMSCAN software. Concerning the At. % (mole %) the total is 100. Authors think that there is a confusion between At.% and At wt..
- Particle size distribution was shown in fig. 9. However, if it can be compared with the particle size analyzer test, it will be more appropriate.
Authors do agree with the reviewer about a comparison with size analyser tests. In this publication, authors just introduce the QEMSCAN database development and illustrate some simulation using physical parameters as particle size or relative density. Particle size analyser will be used for comparison in our future works of BMs characterization.
- The composition and textural information is looks interesting. However, if pure metal, alloy or HEA is forming, can it be analyzed?
The QEMSCAN system is able to distinguish metal alloy and HEA using the peak intensities of EDX spectra.
- How about the gas detection (for example O2) using this methodology?
Unfortunately, gas could not be detected using this methodology

Reviewer 2 Report
This paper deals with the recovery the Black Mass which contains nickel, cobalt, manganese and lithium as valuable elements as well as graphite, solvent, plastics, aluminium and copper. using the combination SEM/QEMSCAN® analyses.
Abstract structure: the content and structure of the abstract is convincingly presented.
Introduction is convincingly presented.
Materials and Methods (Chemical analyses, Sample preparation, QEMSCAN® acquisition parameters), as well as Analytical methodology (Introduction to the QEMSCAN® system, Database development, Methodology, Database verification) and Micro-texture and chemistry of particle are well described.
Conclusions and perspectives corresponds to the conceptual architecture of the paper.
Pictures: adequate number, presentation clearly and good quality.
Adequate list of the literature cited in the references.
In my opinion, revisions are not required - recommendation: Accept
Author Response
Reviewer 2
This paper deals with the recovery the Black Mass which contains nickel, cobalt, manganese and lithium as valuable elements as well as graphite, solvent, plastics, aluminium and copper. using the combination SEM/QEMSCAN® analyses.
Abstract structure: the content and structure of the abstract is convincingly presented.
Introduction is convincingly presented.
Materials and Methods (Chemical analyses, Sample preparation, QEMSCAN® acquisition parameters), as well as Analytical methodology (Introduction to the QEMSCAN® system, Database development, Methodology, Database verification) and Micro-texture and chemistry of particle are well described.
Conclusions and perspectives corresponds to the conceptual architecture of the paper.
Pictures: adequate number, presentation clearly and good quality.
Adequate list of the literature cited in the references.
In my opinion, revisions are not required - recommendation: Accept
Authors are glad to see the report and thank the reviewer for the feedback.

Reviewer 3 Report
The manuscript is a characterization of the Black Mass component of the Lithium-ion batteries using QEMSCAN. The characterization of Libs component by a quantitative mineralogy manuscript is very interesting. However, the characterization of phases in a system with the presence of light elements, such as lithium, cannot be complete by any type of EDS technique and this must be complemented and other techniques, such as X-ray Powder diffraction.
More references are necessary.
Other minor especific comments are:
Line 30. This will not happen in the future but now.
Line 61. These techniques are complementary. To do a phase quantification we need to use a combination of them. Then you should replace “or” by “and”.
Line 66. The first time that you use an acronym you should also write this also in full.
Why you talk about MLA here, may be you can compare with QEMSCAN, that is the equipment that you use here.
It is not TESCAN but TIMA-X
Line 64-70 this paragraph should be rewritten.
Line 76. This is part of your results and should not be mentioned here.
Line 100. “The ratio between the three valuable elements is close to 2:1.“ There are thre elements, then this should be “The ratio among the three valuable elements is close to 2:1: ?“.
Line 105. Repalce “As described by Vanderbruggen et al. [9], the sampling and the representativity of samples are important for automated mineralogical analyses“ by “The sampling and the representativity of 105 samples are important for automated mineralogical analyses [xx, 9].“
Many authors before [9] reported problems about the sample preparation.
Line 129. FEI Quanta 650 FEG SEM (Company, City).
Figure 2. The authors read few articles on phases liberation, there are many other figures in the literature easy to understand that this presented here. In the second image this seems that the foil piece and the lithium phases are close, but this does not mean that they are non-liberated. Liberation means that they are hooked together and cannot be easily separated by physical methods.
Line 204 Silicates ….were
Figure 11. The name of some phases is not indicated.
Figure 16. It should be more interesting to show a smaller part of the image of the concentrate, which allows a better view of the morphology and composition of the different phases.
Line 285, 308 and 342 I think that this should be in the methods section or, at least, you could present these data in a table.
Figure 14. Are the ranges present in colours the particle size classes?. If yes, there are figures where you can do a 3D graph and represent the particle grade % liberation) / particle size classes / mass.
Author Response
The manuscript is a characterization of the Black Mass component of the Lithium-ion batteries using QEMSCAN. The characterization of Libs component by a quantitative mineralogy manuscript is very interesting. However, the characterization of phases in a system with the presence of light elements, such as lithium, cannot be complete by any type of EDS technique and this must be complemented and other techniques, such as X-ray Powder diffraction.
Authors do agree with the fact that lithium can not be detected using EDX technique and this is why authors highlight this in line 212-215.
More references are necessary.
Other minor especific comments are:
Line 30. This will not happen in the future but now.
Authors have modified the sentence
Line 61. These techniques are complementary. To do a phase quantification we need to use a combination of them. Then you should replace “or” by “and”.
Authors take into account the reviewer proposition.
Line 66. The first time that you use an acronym you should also write this also in full.
Why you talk about MLA here, may be you can compare with QEMSCAN, that is the equipment that you use here.
It is not TESCAN but TIMA-X
Authors modified “MLA technology” with “QEMSCAN® technology”. Moreover, authors add “(TIMA-X)” to get information about the TESCAN model SEM.
Line 64-70 this paragraph should be rewritten.
Authors have slightly modified the paragraph line 64-70 but do not fully understand the reviewer comments on completely rewriting this paragraph.
Line 76. This is part of your results and should not be mentioned here.
Authors have modified the sentences.
Line 100. “The ratio between the three valuable elements is close to 2:1.“ There are thre elements, then this should be “The ratio among the three valuable elements is close to 2:1: ?“.
Authors take into account the proposition of the reviewer and add some complementary information.
Line 105. Repalce “As described by Vanderbruggen et al. [9], the sampling and the representativity of samples are important for automated mineralogical analyses“ by “The sampling and the representativity of 105 samples are important for automated mineralogical analyses [xx, 9].“
Many authors before [9] reported problems about the sample preparation.
Authors take into account the proposition of the reviewer without mentioning the 105 samples which could be a tapping fault. However, authors do not quote more publications to this point because this is not the aim of the present paper.
Line 129. FEI Quanta 650 FEG SEM (Company, City).
Authors added information
Figure 2. The authors read few articles on phases liberation, there are many other figures in the literature easy to understand that this presented here. In the second image this seems that the foil piece and the lithium phases are close, but this does not mean that they are non-liberated. Liberation means that they are hooked together and cannot be easily separated by physical methods.
Figure caption were slightly modified. The definition of liberation used in automated mineralogy approach does not address a potential separation of texturally associated phases.
Line 204 Silicates ….were
Authors have rectified the sentence
Figure 11. The name of some phases is not indicated.
Authors have identified the missing phases in the Figure 11
Figure 16. It should be more interesting to show a smaller part of the image of the concentrate, which allows a better view of the morphology and composition of the different phases.
Authors understand the approach of the reviewer; however, it is very difficult to get more information about the morphology using a step width of 5 µm with particles of above 20 µm (4 squares illustrate these ultrafine particles). For future work, authors will use a lower step width (1 µm) to acquire more information on ultrafine particles.
Line 285, 308 and 342 I think that this should be in the methods section or, at least, you could present these data in a table.
Authors understand the proposition of the reviewer however we think that it does not play a significant role for the understanding of the methodology.
Figure 14. Are the ranges present in colours the particle size classes?. If yes, there are figures where you can do a 3D graph and represent the particle grade % liberation) / particle size classes / mass.
The ranges present in colours correspond to the liberation %. Authors know that 3D graphs with liberation %/particle size/mass could also be generated. However, authors try to present an illustrated figure of liberation for each BM which will be easier to understand for future readers.

Round 2
Reviewer 3 Report
The article has been improved respect to the first version, but I consider that still are some aspects that should change.
The objective indicated should be more explicit, it cannot be an objective to identify some phases more than others.
I consider that the section 3: "Analytical methodology should be part of the section 2. "materials and methods"
I still do not agree with including a figure that comes from another article published in another journal. Moreover, this figure is not very illustrative, since the Al foil and the mineral grains are not part of the same particle.
I therefore insist that this figure should be changed, otherwise I cannot recommend the publication of the article. Some articles can hept you to do your own figure, eg.:
Lottering et at. Miner. Eng. 2008, 21, 16-22.
Becker et al. J. Miner. Process. 2009, 93, 246-255
Bushell, Minerals Eng. 2012, 36-38, 75-80.
Jodens et al. Int. J. Minera. Process. 2016, 155, 6-12)
The title of your favorite citation [9] has a mistake "Automated mineralogy as a novel approach for the compositional and textural characterization of spent lithium-ion batteries.
Author Response
The article has been improved respect to the first version, but I consider that still are some aspects that should change.
The objective indicated should be more explicit, it cannot be an objective to identify some phases more than others.
The objectives of this paper (particularly the database development and its application) were more highlighted in the introduction part.
I consider that the section 3: "Analytical methodology should be part of the section 2. "materials and methods"
As suggest, the authors have regrouped the two sections.
I still do not agree with including a figure that comes from another article published in another journal. Moreover, this figure is not very illustrative, since the Al foil and the mineral grains are not part of the same particle.
I therefore insist that this figure should be changed, otherwise I cannot recommend the publication of the article. Some articles can hept you to do your own figure, eg.:
Lottering et at. Miner. Eng. 2008, 21, 16-22.
Becker et al. J. Miner. Process. 2009, 93, 246-255
Bushell, Minerals Eng. 2012, 36-38, 75-80.
Jodens et al. Int. J. Minera. Process. 2016, 155, 6-12)
Authors change the figure with our own figure for liberation explanation
The title of your favorite citation [9] has a mistake "Automated mineralogy as a novel approach for the compositional and textural characterization of spent lithium-ion batteries.
Authors have corrected the fault
